# Fenfluramine: A Review of Pharmacology, Clinical Efficacy, and Safety in Epilepsy

**DOI:** 10.3390/children9081159

**Published:** 2022-08-02

**Authors:** Debopam Samanta

**Affiliations:** Child Neurology Section, Department of Pediatrics, University of Arkansas for Medical Sciences, 1 Children’s Way, Little Rock, AR 72202, USA; dsamanta@uams.edu; Tel.: +1-(434)806-1441; Fax: +1-(501)364-6077

**Keywords:** seizure, epilepsy, refractory, intractable, serotonin, children, pediatrics, AEDs, antiepileptic drug

## Abstract

Despite the availability of more than 30 antiseizure medications (ASMs), the proportion of patients who remain refractory to ASMs remains static. Refractory seizures are almost universal in patients with epileptic encephalopathies. Since many of these patients are not candidates for curative surgery, there is always a need for newer ASMs with better efficacy and safety profile. Recently, the anti-obesity medication fenfluramine (FFA) has been successfully repurposed, and various regulatory agencies approved it for seizures associated with Dravet and Lennox–Gastaut syndromes. However, there is a limited in-depth critical review of FFA to facilitate its optimal use in a clinical context. This narrative review discusses and summarizes the antiseizure mechanism of action of FFA, clinical pharmacology, and clinical studies related to epilepsy, focusing on efficacy and adverse effects.

## 1. Introduction

Epilepsy is one of the most common neurological disorders. Approximately 50 million people globally live with epilepsy. Epilepsy is primarily treated with antiseizure medications (ASMs), and appropriate treatment can make 60–70% of the patients seizure-free [1]. However, the rest of the individuals develop intractable seizures despite receiving proper treatment. In addition, these patients have a higher risk of associated comorbidities and poor health-related quality of life. Unfortunately, the availability of more than 30 ASMs has not changed the proportion of patients who become seizure-free. Thus, there is always a search for novel ASMs that can help reduce seizures and other comorbidities while enhancing the quality of life without causing undue harm. Initial ASMs were developed to modulate voltage-gated sodium channels or enhance GABAergic neurotransmission. However, a number of more recent ASMs have made their way into clinical use. These ASMs, which were either accidentally discovered or developed using animal seizure models, aim to correct the excitation–inhibition imbalance by modulating other neurotransmitter systems, such as the serotonergic system.

### 1.1. The Popularity of Fenfluramine as an Appetite Suppressant

Fenfluramine (FFA), a serotonergic medication, has recently gained popularity as an ASM. It is an amphetamine derivative and racemic mixture of D- and L-enantiomers [2]. It was used in France as an antidepressant and then soon after for weight loss in 1963 [3]. In 1973, it was approved in the USA under the brand name Pondimin™ [3]. However, it never gained widespread acceptance due to side effects, such as nausea, anxiety, and uncomfortable feeling. To increase potency and lessen adverse effects, FFA was combined off-label with another amphetamine derivative, phentermine, in the 1990s [4]. This combination became widely popular as fen-phen. In 1996, the D-enantiomer of FFA, dexfenfluramine (D-FFA), was approved in the USA under the brand name Redux™, a more potent appetite suppressant than FFA [5]. Both FFA and D-FFA, with or without phentermine, became extremely popular weight loss agents [6]. Between 1993 and 1996, more than 50 million prescriptions for these drugs were written in Europe, with 18 million in 1996 alone [7]. Similarly, in the first five months after its approval in 1996, D-FFA was prescribed 1.2 million times in the United States [7].

### 1.2. Fenfluramine’s Downfall as an Appetite Suppressant

FFA and D-FFA rapidly declined from huge popularity as reports of serious cardiovascular adverse effects emerged. In 1981, two cases of pulmonary artery hypertension (PAH) associated with FFA use were reported [8]. Several other sporadic cases of PAH were also reported subsequently [9]. In 1996, a case-control study compared 95 patients with PAH with 355 healthy controls. PAH patients had 23 times higher odds of exposure to three months of anorexic agents, primarily D-FFA [10]. In 1997, a case series reported 24 females who developed mitral and aortic regurgitation following 11 months (range 1–28 months) of fen-phen exposure; five of these patients required valve replacements [11]. The average dose of FFA was 40–60 mg/day (range 20–220 mg/day). More than 100 cases of valvulopathy with a mean exposure of 9 months were reported to the CDC soon after [5]. Approximately 98% of the patients were women, 27 required valve replacements, and three died despite surgery. Based on these severe adverse effects, both FFA and D-FFA were removed from the US market in September 1997 [3].

### 1.3. The Reemergence of Fenfluramine as an Antiseizure Medication

Before FFA was withdrawn globally from the market, it was noted to be effective in pediatric epilepsy in the 1980s and 1990s. Belgian researchers were able to continue using FFA for epilepsy with the government’s help. A Belgium Royal Decree was issued in March 2002 to make it available for treating seizures as the drug stopped being commercially available [6]. Subsequently, more case series were published regarding FFA’s efficacy as an adjunctive treatment option for an orphan indication, namely treating seizures associated with Dravet syndrome. Increased awareness of the unmet needs related to rare epilepsy and regulatory incentive to develop drugs for these indications promoted multiple randomized controlled studies (RCTs) using FFA in Dravet and Lennox–Gastaut syndromes (LGS). In 2020, the FDA and the European Union approved FFA for seizures associated with Dravet syndrome in >2 years old. Subsequently, in March 2022, the FDA approved it for treating seizures associated with LGS.

This narrative review discusses and summarizes the antiseizure mechanism of action of FFA, clinical pharmacology, and clinical studies related to epilepsy, focusing on efficacy and adverse effects. MEDLINE using PubMed and OvidSP vendors were searched in March and June 2022 with appropriate keywords (“Fenfluramine” in combination with “seizures” and “epilepsy”). To find other pertinent publications, the author used a “snowball sampling” approach to go through each article’s references. After reviewing the titles and abstracts of all these articles, relevant English language FFA articles (preclinical and clinical studies) were included in this review. The following standards were used in compiling the final reference list: innovation, importance, quality, and relevance to the review’s objectives.

## 2. The Mechanisms of the Antiseizure Effect of Fenfluramine

FFA controls the serotonergic system by increasing extracellular serotonin (5-hydroxytryptamine/5-HT), which in turn activates 5-HT receptors. First, FFA interacts with serotonin transporter to inhibit 5-HT uptake [12]. In addition, FFA prevents 5-HT from moving to the vesicles from the cytoplasm. Instead, this cytoplasmic 5-HT gets released outside the cell using 5-HT carriers, causing increased extracellular 5-HT levels.

Increased 5-HT levels in the brain were noted to inhibit seizures in rats since 1957 [13]. In experimental conditions, extracellular 5-HT levels increased after exposure to several anticonvulsants, including phenytoin and carbamazepine [14]. Increased 5-HT is also noted with selective serotonin reuptake inhibitors (SSRIs); some of these SSRIs positively affect seizure management [15]. Conversely, decreased CSF 5-HT was associated with increased seizure susceptibility to audiogenic, chemical, and electrical stimulation.

Aside from the indirect effect of increasing extracellular 5-HT levels, FFA’s metabolites, D- and L-norfenfluramine (norFFA), activate 5-HT receptors directly [16,17]. 5-HT receptors are present in the neocortex and hippocampus. Fourteen different 5-HT receptors are classified into seven distinct families (5-HT_1_-5-HT_7_). Except for 5-HT_3_, other 5-HT receptors are G protein-coupled seven-transmembrane receptors that activate the intracellular second messenger cascade. 5-HT_3_ is a ligand-gated Na-K channel responsible for an excitotoxic effect causing depolarization. Although additional research is needed on which specific 5-HT receptor subtypes and underlying pathways are engaged, 5-HT_1D_ and 5-HT_2c_ receptors may be primarily responsible for the antiseizure action of FFA [17]. 5-HT_1_ receptors decrease adenyl cyclase and cyclic adenosine monophosphate, causing inhibitory effects, whereas 5-HT_2_ receptors activate phospholipase C and increase inositol phosphate and diacylglycerol, leading to excitation. In general, increased 5-HT enhances GABAergic neurotransmission in inhibitory interneurons by increasing the frequency and amplitude of spontaneous inhibitory postsynaptic currents [18]. In addition, stimulation of the 5-HT_2A_ and 5-HT_2c_ could inhibit rhythmic thalamic burst firing, resulting in possible effectiveness in absence seizures.

The anticonvulsant effect of FFA is diverse in experimental models. FFA could block epileptiform activity induced by lowering Mg concentration in rat entorhinal cortical slices [19]. FFA’s action could be blocked by a 5-HT_1A_ receptor antagonist or 5-HT reuptake inhibition. In mice, FFA could protect against audiogenic seizures and seizure-induced respiratory arrest [20]. In a maximal electroshock seizure (MES) model, 10 or 20 mg/kg intraperitoneal FFA prevented seizures in rats but not in mice [21]. Similarly, FFA did not prevent chemical (pentylenetetrazole) and electrical stimulation-induced clonic seizures in different animal models (rats, mice, and zebrafish) [20]. In a Zebrafish model, FFA suppressed epileptiform discharges due to 5-HT_1D_ and 5-HT_2C_ agonist activity [22]. FFA could block seizure-like hyperactivity and epileptiform discharges in the antisense knockdown Zebrafish model [23]. In addition, chronic exposure to FFA could reverse the loss of dendritic arbors in GABAergic interneurons if instituted early [24]. Limited research has been done regarding which enantiomer of FFA is most active or essential to prevent seizures. D-enantiomers (FFA and norFFA) reduced seizure-like behavior in a Dravet Zebrafish model by 80–84%, while L-enantiomers (FFA and norFFA) only by 41–48%; nonetheless, both enantiomers lowered epileptiform discharges virtually equally [25].

Beyond serotonergic activity, FFA reacts with other receptors, most notably a subtype of plasma membrane-localized opioid receptor, namely sigma-1 receptors [17,26]. While the serotonergic activity of FFA at the 5-HT receptors primarily increases GABAergic signaling, its sigma-1 receptor activity decreases glutamatergic excitability [27]. These opioid receptors are chaperone proteins that mobilize intracellular Ca^2+^ stores and modulate ion channels and neurotransmitters. For example, sigma-1 receptors may bind to N-methyl-D-aspartate (NMDA) and G-protein coupled receptors to control Ca^2+^ influx. Although initially FFA was suspected to be an antagonist of sigma-1 receptors, later, it was thought to induce positive allosteric modulation, which may have additional benefits on behavior and cognition [28,29,30]. An earlier study showed that D-FFA might have preferential activity as a positive allosteric modulator than L-FFA and D-norFFA [31]. Sigma-1 receptors also interact with neuropeptide Y (NPY). The NPY level is likewise impacted by FFA. Importantly, NPY also has independent anticonvulsant action by decreasing glutamatergic synaptic transmission [32].

## 3. Fenfluramine Clinical Pharmacology

Following quick absorption from the gastrointestinal tract, FFA reaches peak plasma concentration in 3–5 h [33]. Absolute oral bioavailability is 75–83% and is not significantly affected by food intake [5]. Further, >75% of FFA is metabolized (de-ethylated) in the liver to norFFA by six cytochrome P 450 enzymes (primary: CYP1A2, CYP2B6, and CYP2D6; secondary: CYP2C9, CYP2C19, and CYP 3A4) [34,35]. FFA and norFFA are approximately 50% protein-bound. Subsequently, FFA and its metabolites are renally excreted with an elimination half-life of 20 h. NorFFA has a longer half-life of 24–48 h.

Stiripentol acts on all six of the CYP450 enzymes that affect FFA metabolism [36]. Concomitant stiripentol treatment causes significantly higher FFA and lower norFFA levels. Other ASMs, namely clobazam and valproate, which inhibit CYP2C9 and CYP26, respectively, also have an impact on FFA metabolism [37]. Although a higher FFA level is concerning for various side effects, including cardiac toxicity, the lower norFFA level can be protective as norFFA may be the primary culprit causing cardiac toxicity via 5-HT_2B_ receptors. FFA and its active metabolite norFFA weakly inhibit CYP2D6 and induce CYP2B6 and CYP3A4 [38,39]. FFA and norFFA do not induce or inhibit most other cytochrome p450 enzymes, including CYP1A2, CYP2C9, CYP2C19, or CYP3A5, in a clinically meaningful way at doses used for epilepsy treatment [38,39]. In addition, drug transporters (including OCT2 and MATE1) were not inhibited by FFA or norFFA [38,39]. In general, FFA may not affect the metabolism of other ASMs in significant ways. However, further studies are needed to understand FFA’s interaction with other commonly used ASMs, such as lamotrigine, rufinamide, and topiramate.

There is significant inter-and intra-patient variability in the plasma concentration of FFA and its metabolites. The plasma concentration was evaluated in 321 samples obtained from 61 patients (49 patients with Dravet syndrome and seven with LGS) [40]. Lower concentration was reported in younger patients when adjusted to weight, suggesting faster clearance in children. As most recent studies have used a maximum daily dose of 26 mg, children <37 kg might receive higher mg/kg/dose than adults. However, higher clearance in children suggests that they may not be exposed to higher plasma concentrations of FFA or norFFA. Plasma concentration of FFA increases proportionally when the FFA dose increases in the range of 0.35–0.7 mg/kg/d [5].

A high FFA level has been associated with fatigue and sedation but not anorexia or seizure control. Patients with good responsiveness to FFA may improve even with a low plasma concentration of FFA. A plasma level greater than 150–200 µg/L may not provide additional benefit in seizure control. Previous studies suggested that an FFA level of 240–850 µg/L is associated with excessive serotonergic adverse effects, including hyperreflexia, tachycardia, hyperthermia, etc. [41,42,43]. Death has been reported with a level >6500 µg/L [44].

## 4. Clinical Use of FFA in Epilepsy

### 4.1. Earlier Observational Studies

Since the 1980s, neurologists have reported the use of FFA in treating paroxysmal episodes. In 1984, Gastaut reported FFA’s efficacy (30–60 mg/d) in children with self-induced syncope from the Valsalva maneuver [45]. In 1988, Aicardi et al. reported an 11-year-old with absence epilepsy whose self-induced syncope was successfully treated with FFA [46]. Around the same time, FFA’s effect on photosensitive seizures was reported. In 1985, Aicardi and Gastaut described three patients with photosensitive seizures treated with 60 mg/d of FFA [47]. Gastaut and Zifkin, in 1987, used FFA (0.5–1.5 mg/kg/d) in 33 children with intractable epilepsy, and 46% of these patients had ≥50% seizure reduction [48]. One patient with LGS had >2/3rd seizure reduction. In 1996, Boel and Casaer reported adjunctive use of FFA (0.5–1 mg/kg/d) in 11 children with intractable epilepsy and self-induced seizures [49]. Eight of these patients had photo or pattern sensitivity. Among 11 patients, seven achieved seizure freedom, and the remaining four had >75% seizure reduction over the median five years (range 3–8.5 years) of exposure. Adverse effects were mild and transient. The same group reported their experience of using FFA on 22 patients with self-induced seizures in 2002 [50]. Thirteen of these patients had photosensitivity. Ten patients had >90% seizure reduction, and six became completely seizure-free for 1–12 years. One patient had a transient loss of appetite. In 1988, Clemens reported successful treatment of a patient with LGS and self-induced seizure with a combination of bromide and FFA [51].

### 4.2. Dravet Syndrome Observational Studies

In 2012, Ceulemans et al. published a retrospective study of 12 Dravet syndrome patients (five patients were included in Boel and Caesar’s 1996 study) with intractable epilepsy [52]. Two patients discontinued treatment. With a follow-up period of 11 years, adjunctive FFA (0.34 mg/kg/d) in the remaining ten patients led to seven being seizure-free for at least one year. Two patients had asymptomatic cardiac valve thickening, and two patients exposed to combined treatment with topiramate had appetite suppression. This study suggested FFA can be effective beyond self-stimulation and photosensitive seizures. In 2016, 5-year prospective follow-up data (2010–2014) were published [53]. The study provided a standardized evaluation of 10 original patients. The mean treatment duration of FFA was 16.1 years with an average daily dose of 16 mg/d (0.27 mg/kg/d). The study reported that three patients in the cohort remained completely seizure-free throughout the 5-year-study period, and an additional four patients had ≥2-year seizure-free period. In 2011, a prospective study of FFA (maximum dose of 20 mg/d) was started in Belgium. This study included two new patients other than the ten patients from the previous research. Both new patients (1 and 10 years old) had >75% seizure reduction and complete freedom from status epilepticus [6,54]. In 2019, they reported that all original ten patients continued on average 17.5 mg/d of FFA over 20 years (9–30 years). Mild thickening of valve leaflets in three patients remained stable over >5 years. In addition, none of the patients developed dysfunction of heart valves.

### 4.3. Dravet Syndrome Randomized Controlled Trials RCTs

These initial successful studies led to the arrangement of three randomized controlled trials (RCTs) in Dravet syndrome with intractable convulsive seizures (>six seizures in 6 weeks baseline period) (Figure 1). In the first study, 119 patients (2–18 years old) were treated with FFA (0.2 or 0.7 mg/kg/day; maximum dose 26 mg/d) or placebo for 14 weeks [55]. High- and low-dose FFA showed 62.3% (95% CI 47.7–72.8, *p* < 0.0001) and 32.4% (95% CI 6.2–52.3, *p* = 0.0209) placebo-adjusted seizure reduction in monthly convulsive seizures. Fifty percent seizure-responder was 68% (*p* < 0.001) and 38% (*p* < 0.01) in high-and low dose FFA groups compared to 12% in the placebo. A 75% seizure-responder rate was 50% (*p* = 0.0005) and 23% (*p* = 0.0229) in the high and low-dose FFA groups compared to 2% in the placebo. Eight percent of patients were completely seizure-free in both high and low-dose groups compared to none in the placebo group. Additionally, 18% of additional patients in the high dose group had only one seizure over the entire 14 weeks period. Patients exposed to the high dose also required significantly fewer days of rescue medicines.

In a similar international phase 3 trial, 142 patients with Dravet syndrome had a statistically significant reduction of monthly convulsive seizures in both high and low doses of FFA [56]. FFA 0.7 mg/kg/day showed a 64.8% greater reduction in mean monthly convulsive seizures vs. placebo (*p* < 0.0001). Compared to patients in the placebo group, a significantly higher proportion of patients given either dose of FFA experienced a ≥50% (high dose—73%, [*p* < 0.0001] low dose—46% [*p* < 0.001], and placebo—6%) or ≥75% seizure- reduction (high dose—48%, low dose—28%, and placebo—4%) in monthly convulsive seizures. Patients in both FFA groups had significantly longer seizure-free intervals vs. placebo. In addition, efficacy of 0.7 mg/kg/day exceeded 0.2 mg/kg/day for primary and secondary endpoints, suggesting a dose–response relationship. More investigators and caregivers judged FFA than placebo-treated patients as much improved/very much improved. 

In the third Dravet syndrome trial, the FFA dose was 0.4 mg/kg/d (max 17 mg/d) as this trial included patients taking concomitant stiripentol (in combination with clobazam+- valproate), which inhibits the metabolism of FFA [36]. The placebo-adjusted reduction of monthly convulsive seizures was 54% (*p* < 0.001). The 50% seizure-responder rate was also higher in the FFA group (53.5% vs. 4.5%; *p* ≤ 0.001). A >75% reduction in monthly convulsive seizure frequency was noted in 35% (*p* = 0.003) of patients. The longest mean seizure-free interval was approximately 30 days, about three times higher than the placebo. Finally, the open-label extension of the two RCTs showed that a median decrease in convulsive seizure frequency was maintained (−66.8%) over a longer follow-up (median analysis of 256 day-period) [57].

In the above mentioned three independent pivotal RCTs, the efficacy of FFA against convulsive seizures approximated the responder rate at the ≥75% level that was previously noted at the ≥50% level by other ASMs [58]. Furthermore, the number needed to treat (NNT) with FFA to achieve either ≥50 or ≥75% seizure reduction was 2–3, which is much more favorable than other ASMs used in Dravet syndrome (NNT of 4–6) or when ASMs were evaluated in treating other treatment-resistant epilepsies (NNT 8–20) [58]. Although these RCTs provided population-based metrics of FFA’s effectiveness, its effect on day-to-day seizure burden is also robust. Post hoc time to event analysis was done to evaluate how long it takes for patients to experience the same number of seizures compared to a baseline six-week period. The analysis showed that 58–60% of patients never reached baseline seizure numbers over the entire trial period [59]. These patients also experienced significantly more convulsive seizure-free days (22–25 days with high dose FFA vs. 13 days).

### 4.4. Sudden Unexpected Death in Epilepsy (SUDEP)

Dravet syndrome patients have a high risk of Sudden Unexpected Death in Epilepsy (SUDEP) and all-cause mortality. These patients have frequent GTCS and intractable epilepsy, and both are independent major risk factors for SUDEP [60]. FFA’s effect on SUDEP was investigated as it was postulated that FFA could decrease the risk of seizure-induced respiratory arrest. FFA may work through 5-HT4 receptors in the brainstem to prevent apnea [20,61]. Cross et al. reported a post hoc analysis of the SUDEP rate in 732 Dravet syndrome patients receiving FFA [62]. These patients were collected from phase 3 trials, early access programs, and clinical practice in Belgium. Three deaths were identified: 2 were probable, and one cause of death was determined as definite SUDEP. The SUDEP and all-cause mortality rates were calculated as 1.7 per 1000 person-years, much lower than rates reported in a historical study [63]. In that study, the all-cause and SUDEP mortality rate was estimated at 15.8 and 9.3 per 1000 person-years [63]. However, further studies are needed to understand FFA’s impact on SUDEP since the study by Cross et al. included most patients from phase 3 clinical trials with a short duration of exposure to FFA. In addition, patients receive greater attention during clinical trials, which may decrease the SUDEP rate.

### 4.5. Status Epilepticus (SE)

Status epilepticus (SE) is common in Dravet syndrome and one of the most common causes of death [64]. Although limited research has been conducted to evaluate FFA’s effect on SE, its effect on seizure frequency can be rapid [65]. However, only sporadic case reports of FFA exist that report aborting ongoing SE. Specchio et al. reported an 8-year-old boy with nonconvulsive SE who had increased responsiveness after four days of 26 mg/d of FFA treatment [66]. Later follow-up showed better seizure control. Millett et al. used FFA in a 20-year-old female with Dravet syndrome, who presented with super-refractory tonic status epilepticus [67]. She continued to have seizures despite five weeks of pharmacological coma with pentobarbital and ketamine, and various other ASMs. Then she was started on 0.4 mg/kg/day of FFA that subsequently increased to 0.7 mg/kg/d (32.2 mg/d). Within one week of high-dose FFA therapy, she became seizure-free and was able to successfully weaned from anesthetic agents. As she continued on FFA, long-term follow-up demonstrated no further episode of SE and marked improvement in seizure control. She also had improved alertness and language skills.

Aside from Dravet syndrome, FFA has been investigated in patients with LGS in an open-label study and then in one RCT.

### 4.6. LGS Open-Label Study

Lagae et al. reported FFA’s effectiveness in LGS in an open-label prospective phase 2 study [68]. Among 13 patients with LGS (>4 convulsive seizures/month), 10 patients completed 20 weeks of FFA (0.8 mg/kg/d; max 30 mg/d) treatment. The median reduction of seizures was 53% (60% seizure reduction among ten patients who completed the study), and >50% seizure reduction was noted in eight (62%) patients. Long-term efficacy was evaluated in nine patients, and the median reduction of seizures was 58%. In addition, ≥50% seizure reduction was noted in 67% of patients, and ≥75% reduction was achieved in 33% of patients. Decreased appetite was noted in one-third of patients.

### 4.7. LGS RCT

A multicenter, double-blind, placebo-controlled RCT was conducted in 263 LGS patients (2–35 years old) with ≥2 drop seizures/week [69] (Figure 2). The median percentage reduction in frequency of drop seizures was 26.5% (*p* < 0.01) with 0.7-mg/kg/d, 14.2% (*p* = 0.09) with 0.2-mg/kg/d, and 7.6% in the placebo group during 14 weeks of the treatment period. The 50% seizure-responder rate (28 and 25% vs. 10%; *p* = 0.005 and 0.02) was higher in the 0.2 and 0.7-mg/kg/d FFA groups than the placebo. Among all seizure types, generalized tonic-clonic seizures (GTCS; reported in 46% of the patients in the cohort) appeared to be most responsive to FFA. In contrast to an increase of 3.7% in the placebo group, the frequency of GTCS decreased by 45.7% and 58.2% in the 0.7-mg/kg/d and 0.2-mg/kg/d FFA groups, respectively. More patients in the 0.7-mg/kg/d group than in the placebo group had ‘much improved’ or ‘very much improved’ ratings on the Clinical Global Impression of Improvement scale (26% vs. 6%; *p* =.001). Following completion of the phase 3 RCT, 247 patients enrolled for the open-label extension study [70]. Interim analysis at one year showed that 83 patients (33.6%) discontinued taking FFA in the interval. The primary reason for discontinuation was lack of efficacy (22%). However, other patients had sustained improvement with a median 51.8% reduction of drop seizures at one year from the study baseline. Additionally, 51.2% and 25.3% of patients had ≥50% and ≥75% seizure reduction, respectively, after up to 1-year treatment with FFA.

Although FFA is only approved so far for seizures associated with Dravet and LGS, small-scale studies evaluated its efficacy and safety in other epilepsy syndromes.

### 4.8. Other Epilepsy Syndromes

FFA use has been reported in rare epilepsy called Sunflower syndrome. In this syndrome, patients seek a light source, such as the Sun, and wave hands before them with or without staring and eye fluttering. They also may have GTCS. One of the initial epilepsy studies of FFA included one patient who likely had Sunflower syndrome [47]. Recently, FFA was used for three months in an open-label trial in 10 Sunflower syndrome patients (13 ± 4 years) to study its efficacy further [71]. Dose was 0.7 mg/kg/d; maximum 26 mg/d. Eight out of nine patients who completed the study had ≥30% seizure reduction. Notably, hand-waving episodes were reduced ≥ by 70% in six patients. Absence seizures were decreased in two out of three monitored patients, but the effect on GTC seizures could not be assessed explicitly due to infrequent occurrence. Several patients had improvement in EEG abnormalities; spikes/hour decreased from 73.9 to 50.4. Photoparoxysmal response was noted in three patients compared to five patients before treatment. These patients’ full-scale IQ improved from 97.9 to 106.2 (*p* = 0.06) but was not statistically significant. Other cognitive and behavioral scores did not change following treatment. The most common adverse effects were fatigue and loss of appetite, noted in four patients. Aside from Sunflower syndrome, FFA was used in six patients (2–26 years) with CDKL5 deficiency disorder [72]. FFA was administered over 2–9 months at 0.4–0.7 mg/kg/d. Tonic-clonic and tonic seizures decreased by a median of 90% and 50–60%, respectively. In addition, 67% of caregivers reported overall clinical improvement, and four patients had improved pediatric quality of life scores. However, one patient each had lethargy and decreased appetite.

### 4.9. Early Epileptic Encephalopathy (EE) and Developmental and Epileptic Encephalopathy (DEE)

The use of FFA for early infantile EE and DEE has been reported infrequently. Aeby et al. reported a case of inherited homozygous *SCN1B* missense variant (p.Arg85Cys)-linked to early infantile DEE, who was treated with FFA [73]. As this patient’s seizures were refractory to multiple ASMs (valproic acid, topiramate, clobazam, ketogenic diet) with frequent fever-induced SE, adjunctive FFA was started at 0.6 mg/kg/d at 28 months of age. She did not have further SE with a significant reduction of seizure severity over a 2-year follow-up. Her quality of life also improved. However, her motor or cognitive function did not improve.

### 4.10. Non-Seizure Outcomes

Limited research has been done to assess non-seizure outcomes of FFA, including cognitive function, health-related quality of life, and behavior. Lagae et al. reported improved executive function with short-term (14 weeks) use of FFA [55]. Bishop et al. reported improved executive function evaluated by BRIEF2 scoring following one year of FFA treatment in 58 Dravet syndrome patients [74]. Patients with >50–75% seizure reduction had marked improvement in executive function, especially as measured with Cognitive and Emotional regulation indices. It is currently unclear if the positive effect is due to the direct cognitive-enhancing effect of FFA or due to an improvement in the overall seizure burden. It is also unknown if earlier treatment with FFA can improve cognitive outcomes based on FFA’s possible disease-modifying effect on dendritic arborization.

## 5. Adverse Effects

Short-term adverse effects of FFA were extensively studied in multiple RCTs (Table 1) [36,55,56,69]. Common adverse effects are decreased appetite, diarrhea, pyrexia, nasopharyngitis, lethargy, and drowsiness. Serious adverse effects and the need to discontinue FFA for adverse effects were rare in the controlled studies.

Appetite suppression and weight loss should be specially mentioned as FFA was previously used as an anti-obesity medication. FFA may primarily impact appetite by its effect on 5-HT2c at the hypothalamus [75,76]. FFA also inhibits the release and action of neuropeptide Y, which is responsible for food intake [77]. In short-duration controlled studies, weight loss (>7%) over a few months was noted in up to 20% of patients with 0.7 mg/kg/d of FFA [55]. However, Antonio Gil-Nagel reported that 279 patients exposed to FFA over one year and 128 patients exposed over two years had minimal impact on height or weight [78].

Cardiac valvulopathy is a significant concern secondary to FFA exposure [79]. The cardiac effect is primarily mediated via D-nor FFA’s (10 times more potent than l-enantiomer) effect on 5-HT_2B_ receptors [80,81,82]. When FFA was used in high doses for obesity treatment, left-sided heart valves (mitral and aortic) were predominantly affected. This left-predominant valvulopathy may be due to FFA’s effect on the decreased pulmonary clearance of 5-HT, resulting in exposure to higher 5-HT levels on the left-sided valves [83]. 5-HT exposure may cause increased DNA synthesis and progression of cell cycles (activation of phospholipase C, protein kinase c, extracellular regulated kinase) in valve’s interstitial cells [81,84]. These changes lead to valve thickening and increased density of endothelial cells. Besides 5-HT_2B_ stimulation, other contributing factors to cardiac toxicity are 5-HT_2A_ receptor activation and cytoskeletal protein serotonylation [81].

The prevalence and risk factors of valve dysfunction after exposure to high-dose FFA have been studied. The estimated prevalence of valvulopathy depends on the study population and the measurement technique (clinical detection in symptomatic individuals vs. echocardiographic assessment for mild to moderate dysfunction) [85]. A meta-analysis reported an approximately 20-fold higher risk of aortic valve disease and approximately five-fold higher risk of mitral valve disease following exposure to FFA or D-FFA [86]. Later controlled cross-sectional studies reported a much lower prevalence rate, especially when abnormal baseline echocardiogram results and/or preexisting cardiac disease, such as rheumatic fever, were considered. Nevertheless, a high dose has been consistently associated with a higher risk. Although 20–23% of valvulopathy patients were exposed to ≤40 mg/d FFA, 65–71% used ≥60 mg/d [87]. The cardiac risk increased ninefold with an increase in the FFA dose from <40 mg/d to ≥60 mg/d [87]. However, there is no clear understanding of how different plasma levels of FFA and D-FFA affect the risk of valvulopathy. Furthermore, valvulopathy risk is higher in women, the elderly, and those exposed to these agents for a more extended period (7.1 per 10,000 for clinically symptomatic disease with a <4 months exposure vs. 35 per 10,000 with a >4-month exposure) [88,89].

Another adverse cardiovascular effect reported with high-dose FFA was pulmonary arterial hypertension (PAH) [10]. FFA via serotonergic overactivity can induce pulmonary artery smooth muscle cell proliferation, resulting in PAH [90]. The increased serotonergic activity also can vasoconstrict the pulmonary artery. Several other factors have been suggested to play some role in developing PAH, including nitric oxide deficiency [91].

Due to these previously reported adverse effects, cardiovascular adverse effects were monitored extensively in RCTs for Dravet and Lennox-Gastaut syndromes, and no cases of pulmonary arterial hypertension (PAH) or clinically significant signs or symptoms of cardiovascular disease were observed [36,55,56,69]. In addition, none of these cardiovascular adverse events were also noted during the open-label extension phase (median exposure of 23.9 months) in 327 Dravet syndrome patients who were evaluated with serial echocardiography [92].

## 6. Future Perspectives and Conclusions

Multiple rigorous RCTs have shown that FFA is highly efficacious against Dravet syndrome-associated convulsive seizures and, to a lesser degree, against LGS-associated drop seizures. An international consensus on the management of Dravet syndrome noted FFA as a possible first-line therapy with moderate to strong consensus among physicians and caregivers [93]. Both physicians and caregivers rated it as an agent with excellent tolerability. In addition, it was regarded to cause improved alertness and behavior among patients. Despite its positive perception among physicians and caregivers, one of the barriers to its widespread acceptance is fear of cardiovascular adverse effects and the requirement of echocardiography every six months. Although initial data regarding cardiovascular toxicity are reassuring, the safety program, such as REMS (Risk Evaluation and Mitigation Strategy), would help accrue further long-term data and potentially lessen anxiety among caregivers and physicians.

Although FFA’s role in managing Dravet syndrome-associated seizures is receiving strong support from physicians and caregivers, its real-world impact in treating LGS-associated seizures is still unknown. Importantly, there is no standard treatment algorithm for LGS, and there are seven FDA-approved ASMs [94]. In addition, the efficacy of FFA for LGS-associated drop seizures, though comparable to some other FDA-approved ASMs, is less robust than its efficacy against DS-associated convulsive seizures [95]. In the future, head-to-head comparative studies of FFA and other ASMs can help rational selection of specific ASM [96]. Although the underlying rationale behind FFA’s better efficacy in DS is unclear, GTCS associated with LGS also responded better than other seizure types [69]. Future studies should evaluate response rates of various seizure types. In addition, more studies are necessary to understand the FFA’s effect on learning, behavior, and overall quality of life. As available treatments are unlikely to lead to seizure remission in most patients with LGS, greater benefit in cognitive and behavioral outcomes may justify earlier use of FFA in the disease course. Finally, the role of FFA in the treatment algorithm may also be guided by its cost-effectiveness. Weston et al. reported higher cost-effectiveness (or lower incremental cost-effectiveness ratio) of FFA than cannabidiol in patients with Dravet syndrome [97]. Similar studies are needed in patients with LGS and other epilepsies.

The potential efficacy of FFA in epilepsies other than Dravet syndrome and LGS is unknown, although a small case series of other epileptic encephalopathy suggested its potential utility in other epilepsy syndromes [98,99]. A pilot clinical trial is ongoing to evaluate FFA’s efficacy in different developmental and epileptic encephalopathies (DEEs) secondary to SYNGAP1 encephalopathy, STXBP1 encephalopathy with epilepsy, inv-dup (15) encephalopathy, cortical malformation, and encephalopathy associated with continuous spikes and waves during sleep. Besides assessing seizure efficacy over three months, other secondary outcomes in cognition and behavior will be evaluated in this study over 12 months. In addition, a phase 2 clinical trial of FFA will enroll patients with refractory infantile spasms (NCT04289467). These patients will be treated with 0.8 mg/kg/day FFA for 21 days. Patients with favorable responses will have an option to continue treatment for up to 6 months. Finally, some genetic epilepsy syndromes, such as Pitt–Hopkins and Bainbridge–Ropers syndrome, are associated with refractory epilepsy and breath-holding spells [100]. As FFA was reported to have a positive effect on other nonepileptic paroxysmal events, the efficacy of FFA in mitigating both types of paroxysmal episodes can be investigated in these patients.

The antiseizure activity of FFA has been established in some rodent and Zebrafish models. However, further research should evaluate FFA’s impact in standard animal models (amygdala kindling model, absence epilepsy model, post-traumatic model). These animal studies may lead to a better understanding of which epilepsy syndromes FFA would be most effective. Similarly, further studies may focus on FFA and its enantiomers’ effect on specific 5-HT receptors. These studies may be helpful if the enantiomer-specific antiseizure effect of FFA becomes evident. D-enantiomers of either FFA or norFFA are primarily responsible for adverse effects, such as anorexia and cardiac valvulopathy. If l-FFA is found to have a significant independent antiseizure impact, a chiral switch with developing this single enantiomer can be a safer treatment option.

## Figures and Tables

**Figure 1 children-09-01159-f001:**
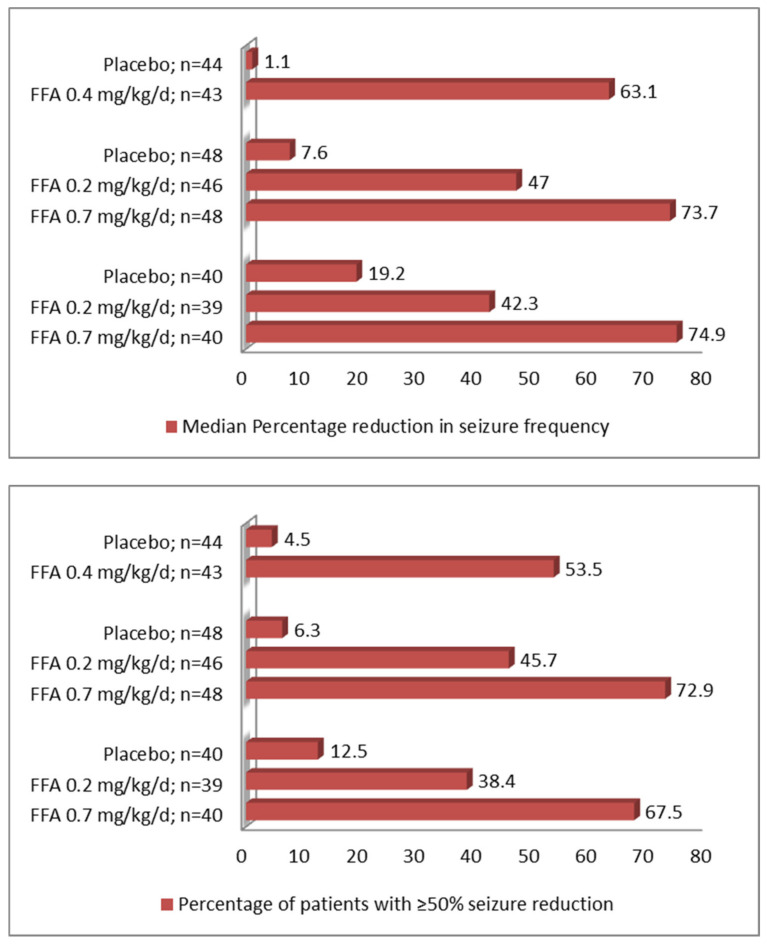
Median percent reduction from baseline in monthly convulsive seizure frequency and ≥50% seizure-responder rates as reported in three randomized controlled trials of fenfluramine in patients with Dravet syndrome (From bottom to up: [36,55,56]) values with statistical comparison were documented in the text.

**Figure 2 children-09-01159-f002:**
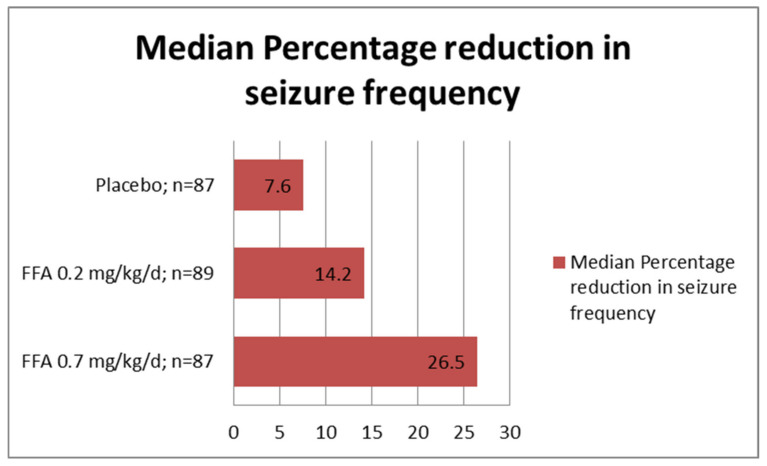
Median percent reduction from baseline in monthly drop seizure frequency and ≥50% seizure-responder rates as reported in the randomized controlled trial of fenfluramine in patients with Lennox–Gastaut syndrome [69]. *p* values with statistical comparison were documented in the text.

**Table 1 children-09-01159-t001:** Reported adverse events of fenfluramine and placebo groups in randomized controlled trials.

	Placebo	Fenfluramine 0.2 mg/kg/day	Fenfluramine 0.4–0.7 mg/kg/day
Reported adverse events	65–83%	78–95%	95%
Decreased appetite	5–11%	20%	36–44%
Diarrhea	7–8%	31%	18–23%
Nasopharyngitis	12–34%	10%	16–18%
Lethargy	5–9%,	10%	18–26%
Somnolence	8%	15%	10%
>7% weight loss	2–4.5%	2–13%	8–20.9%
Echocardiographic finding with trace mitral or aortic regurgitation.	7–13%	18%	23–25%
Pulmonary arterial hypertension or clinically significant signs or symptoms of cardiovascular disease	0	0	0

## Data Availability

Not applicable.

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
