# Peer review of "Fenfluramine: A Review of Pharmacology, Clinical Efficacy, and Safety in Epilepsy"

_children, 2022, doi:10.3390/children9081159_

Round 1

Reviewer 1 Report

This is a very interesting research. The topic taken up is definitely worth a deeper analysis.

Here are some comments:

- I propose to change AMS to AEDs

- the criteria for inclusion and exclusion of publications from the study are missing

- the description of the strategy of searching publications is missing: how many people searched for works, how long did it take to review the publications, what scheme of procedure, how many works were included.

- some references in the bibliography are outdated. it is recommended to use reports not older than 10-15 years.

Author Response

Thank you for reviewing the manuscript entitled ". Fenfluramine: A Review of Pharmacology, Clinical Efficacy and Safety in Epilepsy." I am grateful for your insightful comments Please find below an itemized list of the changes. 

Point 1: - I propose to change AMS to AEDs

Response: Thank you for your thoughtful advice. I concur that antiepileptic drugs (AEDs) were utilized frequently in early medical literature. Despite the widespread use of the term ‘AED’ in the literature, antiseizure medication (ASM) is now the preferred terminology since these agents only symptomatically abort or prevent seizures and have no known effects on epileptogenesis. A citation is included below to justify the use of the terminology. I will include AED (antiepileptic drug) in the keywords to indicate that these are equivalent terms.

French JA, Perucca E. Time to start calling things by their own names? The case for antiseizure medicines. Epilepsy Currents. 2020 Mar;20(2):69-72.

Point 2- the criteria for inclusion and exclusion of publications from the study are missing.

Point 3- the description of the strategy of searching publications is missing: how many people searched for works, how long did it take to review the publications, what scheme of procedure, how many works were included.

Response: We have not used strict inclusion and exclusion criteria for this narrative review, except non-English articles, were excluded. All relevant studies related to antiseizure mechanism of action, clinical pharmacology, and clinical studies related to epilepsy, focusing on efficacy and adverse effects were included in the review stage. English-language publications were found in databases by using different combinations of the following keywords: "Fenfluramine" in connection with "seizures" and "epilepsy." To find other pertinent publications, a "snowball sampling" approach was used to go through each article's references. All of these articles' titles and abstracts were reviewed, and then we thoroughly analyzed the pertinent original studies, including preclinical and clinical studies[ randomized clinical trial,,non-randomized retrospective or prospective studies), meeting abstracts, ongoing clinical trials(only to assess future prospect)]. The following standards were used in compiling the final reference list: innovation, importance, quality, and relevance to the review's objectives.

I have made changes in the revised draft.

Point 4- some references in the bibliography are outdated. it is recommended to use reports not older than 10-15 years.

Response: Thank you for your suggestions. Old references are included to give readers perspectives about historical development related to fenfluramine’s rise in popularity as an antiobesity drug before its rapid downfall due to reports of cardiac adverse effects.

Reviewer 2 Report

The manuscript describes properties of fenfluramine and its clinical application in humans and especially in the control of epilepsy in children. The review is well organized and contains contemporary data on the pharmacokinetics and pharmacodynamics of fenfluramine. Information about its adverse effects are included. Additionally, earlier studies were evaluated and retrospective information was included. The inclusion criteria of the cited papers are described. The study provides information useful for scientists and clinicians and deserves attention. Conclusion reflects the reviewed data and outlines the directions of further investigations. It can be accepted for publication.

Editorial remarks

Line 179: “Above plasma level of 150-200 µ/L additional benefit in seizure control is unlikely.”

Please check again the dimension of the concentration.

Line 180: “level of 240-850 micro/L” – the same remark

Line 182: “level>6500 micro/L” – the same remark

Author Response

Thank you for reviewing the manuscript entitled ". Fenfluramine: A Review of Pharmacology, Clinical Efficacy and Safety in Epilepsy." I am grateful for your kind support of the paper.  Regarding specific corrections, I have checked the plasma levels and edited those dimensions to microgram/L in the revised draft.